# Selection of Polymorphic Patterns Obtained by RAPD-PCR through Qualitative and Quantitative Analyses to Differentiate *Aspergillus fumigatus*

**DOI:** 10.3390/jof8030296

**Published:** 2022-03-13

**Authors:** Omar E. Valencia-Ledezma, Carlos A. Castro-Fuentes, Esperanza Duarte-Escalante, María Guadalupe Frías-De-León, María del Rocío Reyes-Montes

**Affiliations:** 1Unidad de Micología, Facultad de Medicina, Departamento de Microbiología y Parasitología, Universidad Nacional Autónoma de México, Mexico City 04510, Mexico; oevalencia@correo.xoc.uam.mx (O.E.V.-L.); castrofuenca@gmail.com (C.A.C.-F.); dupe@unam.mx (E.D.-E.); 2Hospital Regional de Alta Especialidad de Ixtapaluca, Carretera Federal México-Puebla Km. 34.5, Pueblo de Zoquiapan, Ixtapaluca 56530, Mexico; magpefrias@gmail.com

**Keywords:** *Aspergillus*, RAPD-PCR, logistic regression, quantitative analysis

## Abstract

The objective of this work was to use the random amplification of the polymorphic DNA–polymerase chain reaction (RAPD-PCR) technique to select polymorphic patterns through qualitative and quantitative analyses to differentiate the species *A. flavus*, *A. fumigatus*, *A. niger* and *A. tubingensis*. Twenty-seven *Aspergillus* isolates from different species were typified using phenotypic (macro- and micromorphology) and genotypic (partial *BenA* gene sequencing) methods. Thirty-four primers were used to obtain polymorphic patterns, and with these a qualitative analysis was performed to select the primers that presented species-specific patterns to distinguish each species. For the quantitative selection, a database was built from the polymorphic patterns and used for the construction of logistic regression models; later, the model that presented the highest value of sensitivity against specificity was evaluated through ROC curves. The qualitative selection showed that the primers OPA-19, P54, 1253 and OPA-02 could differentiate the species. A quantitative analysis was carried out through logistic regression, whereby a species-specific correlation of sensitivity and specificity greater than 90% was obtained for the primers: OPC-06 with a 96.32% match to *A. flavus*; OPF-01 with a 100% match to *A. fumigatus*; OPG-13 with a 98.01% match to *A. tubingensis;* and OPF-07 with a 99.71% match to *A. niger*. The primer OPF-01 discriminated the four species as well as closely related species. The quantitative methods using the selected primers allowed discrimination between species and showed their usefulness for genotyping some of the species of medical relevance belonging to the genus *Aspergillus*.

## 1. Introduction

Within the genus *Aspergillus*, more than 400 species are recognized [1], and the most important ones from a pathogenic point of view are included in the sections *Fumigati*, *Flavi* and *Nigri*, representing more than 95% of the pathogenic *Aspergillus* species. It is important to mention that, in the last two decades, there has been a significant increase in invasive fungal infections, particularly invasive aspergillosis (IA) [2]. Among the most relevant opportunistic pathogenic species within the *Fumigati* section are *A. fumigatus*, *A. lentulus*, *A. fumigatiaffinis*, *A. fumisynnematus*, *A. novofumigatus* and *A. laciniosa* [3]. Due to the life-threatening nature of these infections and their drug resistance, accurate identification of *Aspergillus* spp. is necessary [4,5]. The diagnosis of invasive aspergillosis is challenging, particularly in immunocompromised patients [6]. The signs and symptoms are nonspecific, colonization is difficult to distinguish from an invasive disease, blood cultures are commonly negative and patients are often unable to undergo invasive diagnostic procedures [7]. Culture and microscopic examination remain the “gold standard” but lack sensitivity, while immunoassays with antigens such as galactomannan and glucan detection systems are frequently used but vary in sensitivity and specificity depending on the patient population involved [6,7]. On the other hand, molecular tests have been proven to be useful. However, they have not yet been clinically validated. Due to these discrepancies, [8] suggested using a polyphasic approach as the standard for the identification—that is, using a set of criteria, including morphological characterization, physiological tests, ecological data, extrolite analysis and DNA sequences. In particular, with regard to DNA sequences, the use of multi-locus sequences, such as *β-tubulin* (*BenA*), calmodulin (*CaM*), actin (*Act*) or internal transcribed spacers (ITS) sequences, has been suggested. Comparative sequence analysis of the ITS regions, specifically the non-coding regions ITS1 and ITS2 flanking the 5.8S ribosomal DNA, allows the identification of *Aspergillus* at the subgenus/section level, while the amplification and sequencing of some genes, such as actin, calmodulin, rodlet A and/or *β-tubulin*, allow the distinction of *A. fumigatus* from related species within the same section [9]. However, these methodologies are time-consuming, labor-intensive and, therefore, impractical for most clinical laboratories and can be very expensive when used to study collections of large numbers of isolates. To solve this problem, various methodologies have been used for the identification of *Aspergillus* spp. One of these tools is the random amplification of polymorphic DNA (RAPD-PCR), which has been used with good results in the identification and typing of pathogenic fungi since it has proven to be a fast, simple, low-cost and discriminatory method [10,11,12,13]. [11] re-evaluated the identification of species from a collection of strains of *A. fumigatus sensu lato* to identify atypical species based on the new concept of species delimitation. These authors used the RAPD-PCR technique and sequence analysis of the *β-tubulin* gene, which identified *A. fumigatus sensu stricto; A. lentulus; A. viridinutans; Neosartorya udagawae* and *N*. cf. *nishimurae*. Likewise, [13] identified environmental isolates of *Aspergillus* spp. in Tehran through phenotypic methods and RAPD-PCR, and the identified species were *A. niger; A. flavus; A. tubingensis; A. japonicus; A. ochraceus; A. nidulans; A. amstelodami; A. oryzae; A. terreus; A. versicolor; A. flavipes* and *A. fumigatus*. On the other hand, isolates of *A. fumigatus sensu lato* obtained from raw cows’ milk using morphological and molecular techniques (RAPD-PCR and sequencing of the *BenA* gene) were identified and their genetic variability was estimated [14]. The species identified were *A. novofumigatus; A. fumigatiaffinis*; *A. udagawae; A. lentulus* and *A. fumigatus*, the latter being the most abundant and the one that presented intraspecific variability. Similarly, these techniques have also been used to identify species of other pathogenic fungi, such as *Candida* [15] and *Sphorotrix* [16]. This background indicates that RAPD-PCR may be useful in differentiating *Aspergillus* spp.

The increase in immunocompromised patients associated with invasive aspergillosis, as well as new *Aspergillus* spp., points to the need to identify the causative agents at the species level, and a rapid and reliable diagnosis is necessary, so the objective of this work was to select polymorphic patterns obtained by RAPD-PCR through qualitative and quantitative analyses to differentiate the species *A. flavus*, *A. fumigatus*, *A. niger* and *A. tubingensis*, which are the species of medical importance.

## 2. Materials and Methods

### 2.1. Fungal Isolates

Twenty-seven fungal isolates corresponding to the sections *Flavi*, *Fumigati* and *Nigri* were used in addition to five reference strains from the American Type Culture Collection (Manassas, VA, USA) (Table 1), and fifteen partial sequences of the gene *BenA* obtained from GenBank were used. All isolates were obtained from the collection of the Molecular Mycology Laboratory, Department of Microbiology and Parasitology, Faculty of Medicine, National Autonomous University of Mexico (UNAM). All isolates were cultured on potato dextrose agar (PDA) plates (Bioxon, CDMX, MX) at 28 °C for 4–7 days and identified according to their macro- and micromorphological characteristics. The macro- and micromorphological identification was carried out as previously described [8,17]. Micromorphological identification was carried out through microcultures [18], and the microscopic characteristics of the isolates were recorded using a digital camera.

### 2.2. Monosporic Cultures

From each isolate grown in PDA (Bioxon) for 4–7 days at 28 °C, a conidia suspension was prepared with 1 mL of phosphate buffer (pH 7.4) and 0.05% Tween 20 (PBST). This suspension was diluted 1:1000 with PBST, and 50 μL was taken and inoculated in Petri dishes with PDA (Bioxon), which were incubated at 28 °C. An isolated colony was selected from each plate and cultured in a tube with the same medium at 28 °C. The conidia of the monosporic cultures were conserved in sterile water at 4 °C.

### 2.3. DNA Extraction

From each monosporic culture of *Aspergillus* spp. seeded in PDA (Bioxon), a conidial suspension was obtained that was inoculated in tubes with 50 mL of YEPG culture medium (1% yeast extract, 2% peptone, 2% dextrose) and incubated at 37 °C in an orbital shaker for three days until mycelial growth was observed. The mycelial biomass of each isolate was harvested by filtration and frozen at −20 °C until use. Fungal DNA was extracted using a DNeasy^®^ Plant Mini Kit (Qiagen, Austin, TX, USA). Total extracted DNA was quantified by 1% agarose gel electrophoresis and compared with different concentrations (10, 30 and 50 ng/µL) of phage λ (Gibco BRL^®^, San Francisco, CA, USA) stained with GelRed ™ nucleic acid gel stain 10,000× by Biotium (Fremont, CA, USA). Furthermore, it was also quantified by UV spectrophotometry using a NanoDrop 2000 spectrophotometer (Thermo Fisher Scientific, Waltham, MA, USA).

### 2.4. Amplification of the Partial Sequence of the BenA Gene

PCR amplification of the *BenA* gene was carried out as described by [19], and the oligonucleotides used were *Bt2a* (5′-GGTAACCAAATCGGTGCTGCTTTC-3′) and *Bt2b* (5′-ACCCTCAGTGTAGTGACCCTTGGC-3′). A volume of 25 µL was used for the reaction mixture, with 10 mM of MgCl2, 100 µM of each dNTP, 1U/µL *Taq* DNA polymerase, 10 µM of each primer and 20 ng/µL of DNA. Amplification was carried out in a thermocycler Bio-Rad (Hercules, CA, USA) with the following conditions: 95 °C for 8 min; 35 cycles of 95 °C for 15 s; 55 °C for 20 s and 72 °C for 1 min and a cycle of 72 °C for 5 min. The PCR products were sent for sequencing in both directions by Macrogen USA (Rockville, MD, USA), using the Sanger method.

### 2.5. Sequence Analysis

For the analysis of the sequences, the BioEdit program ver. 7.1.9 was used (www.mbio.ncsu.edu/BioEdit/bioedit.html, 10 May 2021), which allowed us to manually corroborate the sequences obtained in the sequencing process (forward and reverse) of each sample and generate a consensus sequence. Each sequence was analyzed with the Basic Local Alignment Search Tool (BLAST) program [20] (www.blast.ncbi.nlm.nih.gov/blast.cgi, 10 May 2021) to confirm its identity. Subsequently, the sequences were aligned with the MAFFT program (http://mafft.cbrc.jp/alignment/server/, 10 May 2021) [21], and the best evolutionary model applied to this alignment was chosen with the JModelTest 2 program (www.github.com./ddarriba/jmodeltest2, 10 May 2021) [22].

### 2.6. Phylogenetic Analysis

Phylogenetic analysis of the sequences was carried out using the maximum likelihood method. The support values of the internal branches were evaluated by a bootstrap method with 1000 repetitions (values equal to or greater than 70% were considered significant) and the GTR + G + I evolutionary model; the nearest neighbor interchange (NNI) heuristic method was applied. A maximum likelihood (ML) analysis was performed with the MEGA software v.10.1.7 [23]. Reference sequences obtained from GenBank were included in the phylogenetic analysis (Table 2).

### 2.7. RAPD-PCR

The RAPD-PCR method was used following two variants: with a primer according to [24,25] and with a double primer according to [26]. Thirty-four primers were tested (Table 3), as described below. Briefly, the RAPD-PCR was carried out in a 25-microliter volume containing 1X buffer, 2.5 mM MgCl_2_, 20 ng of DNA, a 200 μM concentration of each dNTP Applied Biosystems Inc. (Waltham, MA, USA), 1 U of *Taq* DNA polymerase Invitrogen (Carlsbad, CA, USA) and 100 pmol/μL of each primer. The PCR conditions were as follows: 1 cycle of 7 min at 94 °C; followed by 45 cycles of 1 min at 92 °C; 1 min at 35 °C and 1 min at 72 °C and a final extension of 5 min at 72 °C. The products were subjected to 1.5% agarose gel electrophoresis and then stained with GelRed ™ 10,000× Biotium (Fremont, CA, USA); DNA ladders of 100 and 1000 bp (molecular size marker) Invitrogen (Carlsbad, CA, USA) were used. Images of the gels were captured in a Synoptics Photodocumenter Syngene (Frederick, MD, USA).

### 2.8. Qualitative and Quantitative Analyses of Polymorphic Patterns Generated by RAPD-PCR

#### 2.8.1. Qualitative Selection

Species-specific polymorphic patterns were selected for each of the six species studied by RAPD-PCR with the 34 primers. For the selection, the following inclusion criteria were taken into account: number of bands (more than three bands); molecular size (>500 bp) and the definition of the pattern presented by each of the isolates of the studied species.

#### 2.8.2. Quantitative Selection

From the qualitative analysis of the polymorphic patterns, a database was built considering the number of bands per isolate, the molecular size (bp per band) and the intensity of each band in the following ranges: 0.5 (very faint); 1.0 (faint); 2.0 (intense) and 3.0 (very intense). The database obtained was used for the construction of logistic regression models, considering the number of bands, molecular size, intensity and their interaction as dependent variables and the species of the fungus for each of the 34 primers as the independent variable, using the JMP^®^Pro 13 program (SAS Institute Inc., Cary, NC, USA, for Windows). Subsequently, the significance of the models and the study variables was evaluated to select the model that presented the highest value of sensitivity vs. specificity, represented by graphs of receiver operating characteristic (ROC) curves [27,28]. Using the results of the curves, the primers that generated the band patterns were selected, which had areas greater than 0.9 of selectivity vs. specificity for each of the evaluated species (*A. flavus*, *A. fumigatus*, *A. niger* and *A. tubingensis*). The model obtained from the selected primers generated a mathematical equation that allowed the estimation of the most probable species.

### 2.9. Application of Mathematical Models to Identify Aspergillus spp.

To corroborate the identification of the *Aspergillus* spp., polymorphic patterns were obtained by RAPD-PCR with other isolates of different geographical origin and the primers OPC-06, OPF-01, OPF-07 and OPG-13. Information on the number of bands, molecular size and intensity was obtained to compare these data with those previously obtained for the construction of the logistic regression model and obtain the correlation data of association to the most probable species.

## 3. Results

### 3.1. Macro- and Micromorphology

The isolates used in this work presented the typical macro- and micromorphology described for the *Aspergillus* genus. In terms of macromorphology, for the *Fumigati* section, velvety colonies with a green-blue color (front) and cream color (back) were identified; for the *Nigri* section, dark brown to black, powdery (front) and cream-yellow (back) colonies were observed. Greenish to yellowish colonies were identified in the *Flavi* section. Regarding the micromorphology, for the *Fumigati* section bluish-green conidia without metula, spatulate vesicles and smooth conidiophores were observed; the average size of the conidia was in the range of 1.7–2.1 µm, while the length and width of the vesicles were 15.1–24.1 and 12.8–18.4 µm, respectively. The *Nigri* section presented dark brown to black conidia, occasionally metula, globose vesicles and smooth conidiophores; the length and width of the vesicles were 30.77–51.79 and 31.54–51.28 μm, respectively, while the diameter of the conidia was in the range of 2.20–4.23 μm. Finally, the *Flavi* section presented green conidia, brown globose vesicles and rough conidiophores; the diameter of the conidia was 2.78–4.88 μm, and the length and width of the vesicles were 25.64–31.79 and 27.95–31.79 μm, respectively (Appendix A).

### 3.2. Phylogenetic Analysis

A phylogenetic tree was constructed through the maximum likelihood method; three groups were formed. Group I included two subgroups: subgroup Ia included isolates 146A; 220A; 281C and A28 and ATCC reference strain 1004 corresponding to *A. tubingensis*, which were associated with reference strains of *A. tubingensis* (MT410083.1, MT410082.1 and MT410084.1) with a bootstrap of 91%. Subgroup Ib included the reference strain WB326 of *A. niger* (ATCC) and the isolates 335B, 88C, 387A and 39A, which were grouped with the reference strains of *A. niger* (MT410061.1 and MT410063.1) with a bootstrap of 94%. Group II included isolates 87A, 370B, A15 and 323C and the *A. flavus* strain 9643D-2 (ATCC); they were associated with reference strains of *A. flavus* (MT347712.1, MT347711.1 and MT347713.1) with a bootstrap of 100%. Finally, group III formed two subgroups. Subgroup IIIa was formed with the *A. lentulus* strain MYA3566 (ATCC) and reference strains of *A. lentulus* (MN275501.1, MN275504.1 and MN257703.1) with a bootstrap of 90%. Subgroup IIIb included the isolates MM12, MM16, MM18 and MM21 and the reference strains of *A. fumigatus* (MT347703.1, MT347702.1 and NT347701.1) with a bootstrap of 100% (Figure 1).

### 3.3. RAPD-PCR

The polymorphic patterns generated with each of the 34 primers included in the study showed a greater number of polymorphic haplotypes generated with the primers 1253; OPG-01; OPG-03; OPM-12; OPH-18; OPF-01; OPF-09; P54; P160; PELF; OPM-12; T3B and OPA-02, followed by the primers 1281; 1283; OPA-08; OPC-07; OPG-13; OPG-15; OPC-06; OPG-05; OPH-03; OPG-07; OPF-07; OPE-02 and OPA-03. Meanwhile, the primers that generated a lower number of haplotypes were R108; OPA-19; OPA-16; OPB-12; OPA-17; OPA-15; B04 and OPH-17 (Appendix A (1–43)).

### 3.4. Qualitative Selection

Based on the reproducibility of the patterns and considering the inclusion criteria (number of bands, molecular size and definition of the pattern obtained) for each of the species, the species-specific primers were identified (Figure 2), as shown in Table 4.

The primer OPA-19 generated a unique pattern for *A. fumigatus*, while, for the rest of the species, the patterns obtained with this same primer showed a lower number of bands, as in the case of *A. tubingensis*, which presented two bands of 500 and 1600 bp.

The primer 1253 generated a unique pattern for *A. niger*, while the pattern of bands generated for *A. fumigatus* showed intense bands with molecular sizes of 650, 900 and 1000 bp. On the other hand, the pattern generated for *A. flavus* showed a band pattern similar to that of *A. niger*. However, the characteristic bands of the *A. flavus* pattern were between 400 and 500 bp in size.

The primer OPA-02 generated different polymorphic patterns for all the isolates of *A. tubingensis*; however, it generated unique patterns for the species *A. flavus*, *A. fumigatus* and *A. niger*, which were characterized by intense bands and molecular sizes of 190–2300, 220–2400 and 200–2000 bp, respectively, showing the usefulness of this primer to discriminate the three species.

The primer P54 generated unique polymorphic patterns for *A. flavus*, *A. niger*, *A. fumigatus* and *A. tubingensis*; however, the pattern of *A. fumigatus* was the most defined.

### 3.5. Quantitative Selection

From the database obtained, which included 15,000 pieces of data (number of bands, molecular size and intensity of each band), and from the models generated with the 34 primers, ROC curves were constructed, and based on the results of the species-specific correlation values > 0.95, the following primers were selected, OPC-06, OPF-01, OPF-07 and OPG-13.

Figure 3 shows the results of the ROC curves obtained for the selected primers, where a species-specific correlation was observed with areas greater than 0.95 for the following primers: OPC-06, with an area of 0.9632 for *A. flavus*; OPF-01, with an area of 1.0000 for *A. fumigatus*; OPF-07, with an area of 0.9971 for *A. niger* and OPG-13, with an area of 0.9810 for *A. tubingensis*.

### 3.6. Qualitative and Quantitative Comparison of the Selected Primers

To carry out the qualitative and quantitative comparison between the selected isolates, a sensitivity vs. specificity graph was constructed based on the data obtained from the ROC curves. The graph shows that the percentages of sensitivity vs. specificity vary regarding the qualitatively selected primers compared with those selected quantitatively. The following values were obtained for each of the evaluated species: for *A. flavus*, the primer P54 showed 91.58%, while for OPC-06, 96.32% was obtained; for *A. fumigatus*, the primer OPA-19 presented 75.42% against the primer OPF-01, which was 100%; for *A. niger*, the primer 1253 showed 45.45%, while the primer OPF-07 obtained 99.71% and finally, for *A. tubingensis*, the primer OPA-02 presented 66.07%, while for the primer OPG-13, 98.10% was obtained. Figure 4 shows the percentages of sensitivity vs. specificity of the qualitatively and quantitatively selected primers concerning the *Aspergillus* spp. evaluated.

Figure 4 shows that the qualitatively selected primers showed a lower percentage of sensitivity (ROC area) against specificity compared to those selected quantitatively through logistic regression analysis. This same analysis revealed that the variables used (number of bands, molecular size and intensity of each band) and total bands have a different effect on the mathematical models and on the 34 different primers analyzed. That is, it is capable of making a statistical difference based on this analysis, which allows a more precise selection of the useful primers to identify the different species studied. In addition, the probability value obtained in each of the evaluated polymorphic patterns was statistically significant at *p* < 0.0001. Likewise, to corroborate these results, the corresponding probability formulas were obtained for each primer selected to identify other *Aspergillus* isolates of different geographical origin and assign the most probable species (Appendix A (1–15)).

### 3.7. Application of Mathematical Models to Identify Aspergillus spp.

Appendix A (1–3) shows the results obtained from the logistic regression model using other *Aspergillus* isolates of different geographical origin, which showed 100% correspondence between the primers OPC-06, OPF-01 and OPG-13, and the previously identified species (*A. flavus*, *A. fumigatus* and *A. tubingensis*).

The strongest associations were obtained with the following primers: OPC-06 for *A. flavus* with an association correlation of >0.82; OPF-01 for *A. fumigatus* with >0.99, OPG-13 for *A. tubingensis* with >0.96 and the primer OPF-07 for *A. niger*. The correlation results to identify these species were not consistent regarding the areas obtained in this model; the percentage of sensitivity against specificity was 98.9% for the isolates of *A. fumigatus* and 99.71% for the isolates of *A. niger,* generating conflict in the assignment of species. In addition, *A. niger* showed a high polymorphism with most of the primers used and a low specificity.

The primer OPC-06 allowed for obtaining defined patterns only for *A. flavus* and *A. fumigatus*. However, the intensity of the bands was much higher in the *A. fumigatus* isolates (450, 800, 1100 and 1700 bp). With the primer OPF-01 for *A. fumigatus*, nine bands were obtained, whose molecular sizes were in the range of 300–1900 bp (Figure 5). On the other hand, *A. fumigatus* and *A. flavus* showed very similar polymorphic patterns, with one of the differences being that the *A. fumigatus* pattern showed bands of 1700 and 1900 bp, but they were absent in *A. flavus*. Furthermore, *A. niger* and *A. tubingensis* differed in the number and intensity of the polymorphic bands that they presented. Thus, the primer OPF-01 can be considered useful to discriminate the four species.

In the case of the primer OPF-07, an absence of bands was noticeable in the isolates of *A. flavus, A. niger* and *A. tubingensis*. However, *A. fumigatus* showed a distinctive pattern due to the presence of an intense 650-bp band in all isolates of the species. Therefore, this primer was not considered to have a high discriminatory power. The primer OPG-13, like OPF-07, presented a poor polymorphic pattern for most of the *A. tubingensis* isolates. The isolates of *A. fumigatus* presented a variable polymorphic pattern in the number and intensity of polymorphic bands.

## 4. Discussion

In this study, the RAPD-PCR technique was used to obtain polymorphic patterns and, based on these, perform qualitative and quantitative selections that would allow the identification of specific primers for the species *A. flavus, A. fumigatus* and *A. tubingensis*. The isolates included in the study were identified by macro- and micromorphology at the section level, and, through the sequencing of the partial region of the *BenA* gene and phylogenetic analysis, the identification of the species was confirmed.

The partial sequence of the *BenA* gene has proven useful for studies of the phylogenetic relationships of *Aspergillus* and related species [29]. In addition, the *BenA* marker has been used in other works to identify species within the genus *Aspergillus*, which allows a comparative analysis [30]. Ref. [8] have used other genes for the identification of *Aspergillus* spp., such as *CaM* and the ITS region, although the ITS region has proved to be useful only for identification at the genus/section level. In the present work, the phylogenetic analysis with the *BenA* gene sequences could identify the isolates included in the study among the species *A. flavus, A. fumigatus*, *A. niger* and *A. tubingensis* (Figure 1).

On the other hand, fungal typing through RAPD-PCR offers the best potential in terms of obtaining discriminatory data quickly, easily and profitably. This technique has been used for a wide variety of species and has proven to be fast, easy and reproducible when the conditions and reagents of the RAPD-PCR are kept under control [31]. In this work, the RAPD-PCR protocol was carried out under strict conditions so that the tests were reproducible, and the handling of reagents, equipment and areas was always the same to ensure consistent results.

The number of primers used in genotypic variability studies is variable. For example, Ref. [32] used 100 primers to determine that the virulence of *A. fumigatus* is variable from one strain to another. However, Ref. [33] mentioned that as the number of primers increases, the interpretation of the profiles and the control of the reaction conditions become more difficult, which reduces the reproducibility. Therefore, in this study, 34 primers were used to obtain the polymorphic patterns, which was considered sufficient for the analysis.

Initially, the analysis of the polymorphic patterns was carried out qualitatively, by selecting only three primers per species and taking those that provided defined patterns with a high number of bands and species’ specificity. Among the selected qualitative primers, the most representative ones for the polymorphic patterns generated were OPA-19 for *A. fumigatus* and P54 for *A. flavus*, which also presented a specific pattern for all the species studied. The aforementioned agrees with the results reported by [13] when using the P54 primer and obtaining polymorphic patterns for the typing of 38 *Aspergillus* isolates from air samples in Tehran, Iran. In addition, this and other primers used in this study showed high discriminatory capacity and showed wide genetic diversity among the collected isolates.

The polymorphic patterns obtained for *A. tubingensis* and *A. niger* with the primer 1253 are similar, probably because they are species that belong to the same section (*Nigri*); however, the band pattern of *A. tubingensis* presents a band of 1700 bp, which is absent in *A. niger*, so this primer allows us to differentiate between the two species. Thus, the quantitative selection of the primers that generated specific polymorphic patterns for each *Aspergillus* sp. was useful. In their study, Ref. [34] carried out a quantitative analysis of polymorphic patterns generated by RAPD-PCR and statistically compared data obtained from the amplification, taking the values of each of the bands based on their molecular weights (high, medium and low) to weigh the results and thus eliminate the ambiguity that is generated in the qualitative analysis. Thus, taking these parameters into account, in this work a logistic regression analysis was carried out, which gave us a result of species vs. primer association which was selective and specific with the numerical function and thus useful to discriminate between species of the same genus. Furthermore, the database was validated with the statistical program JMP^®^Pro 13 and was built through visual analysis without using programs such as GelCompar II [35], PyElph [36], Quantity One [37] and Image Master [38], among others, which minimize the bias generated when obtaining data from polymorphic patterns; the data were also not normalized, as recommended in the literature [39]. The logistic regression model allowed the obtaining of ROC curves, which evaluated the sensitivity against the specificity of each primer for the different *Aspergillus* spp. evaluated. From the graphs obtained, the primers that presented an area greater than 0.9 were selected. In turn, it was possible to differentiate with those primers that were qualitatively selected in which lower percentages were obtained, as is the case of the primers OPA-19 with 75.42% and OPA-02 with 66.07% sensitivity against specificity compared to 100% obtained in the primer OPF-01, which was selected as a result of the logistic regression. This evidenced that the selection of primers without operative characteristics does not allow adequate identification.

Additionally, the application of the logistic regression model was evaluated using isolates of different geographical origins for their identification. It was observed that the primers OPC-06, OPF-01 and OPG-13 identified and differentiated previously defined isolates at the species level, which corroborated the ability of the models to discriminate between the species *A. flavus, A. fumigatus* and *A. tubingensis*. Regarding *A. niger*, the models of each primer were not able to discriminate this species due to factors that had a high weight in the statistical model, including the intensity and the similarity of banding with other species. Therefore, the use of other primers was suggested, such as those proposed by [40] that were able to distinguish between species of the same section.

In the group of primers selected quantitatively, OPF-01 showed a discriminatory capacity for the four species used in this work, with a particular discriminatory ability between closely related species such as *A. fumigatus* and *A. lentulus*, since the two belonged to the same section. The difference between these species was that the number of bands obtained for *A. fumigatus* was nine bands with molecular weights between 500 and 1800 bp, while for *A. lentulus*, there were only six bands, with molecular sizes between 300 and 1600 bp. It is worth mentioning that the use of the primer OPF-01 for RAPD-PCR has been reported for other species, such as *Paracoccidioides brasiliensis* [41], but does not allow discrimination between species. These results demonstrate that the application of a quantitative model for the selection of useful primers in the identification of *Aspergillus* spp. allows a more appropriate level of discrimination than commonly used qualitative analysis. This is the first proposal to achieve a methodology of quantitative selection.

## 5. Conclusions

This work shows that quantitative methods, rather than qualitative ones, are useful to select species-specific primers to discriminate between species.

The use of a quantitative method, namely logistic regression analysis, for the selection of primers, used in this study to generate polymorphic patterns through RAPD-PCR, allowed the identification of *Aspergillus* spp. The primers OPC-06, OPF-01 and OPG-13 generated specific polymorphic patterns for *A. flavus, A. fumigatus* and *A. tubingensis*, respectively, which, due to their operational characteristics, were specific and selective for each species. This analysis is the first to achieve a quantitative evaluation with mathematical models to determine the primer species-specific association.

The RAPD-PCR technique allowed the identification of *Aspergillus* spp. quickly and easily, without the need for sequencing methods, which can be very useful to improve the diagnostic methods of aspergillosis.

## Figures and Tables

**Figure 1 jof-08-00296-f001:**
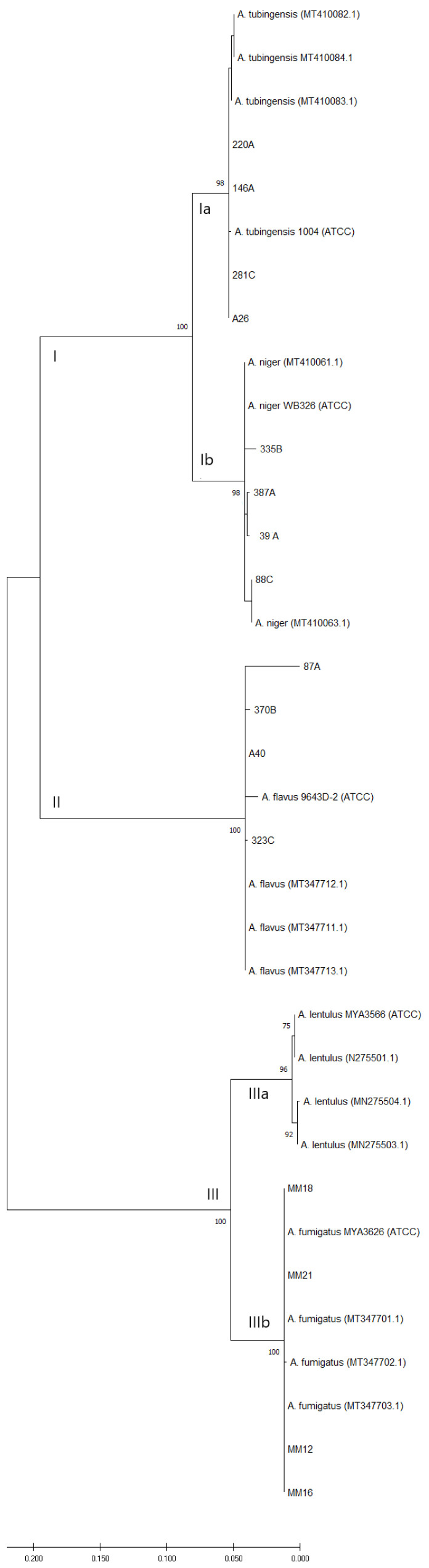
Phylogenetic tree based on the *BenA* gene sequence. The maximum likelihood analysis (ML) was performed with MEGA software v.10.1.7, using the substitution model general time reversible (GTR) model and gamma distributed (+G) with invariant sites (+I) (= GTR + G + I). All positions containing gaps and missing data were included for analysis. Clade supports were calculated based on 1000 bootstrap re-samplings. This analysis involved 35 taxa. The sequences analyzed were in the range of 439–566 bp.

**Figure 2 jof-08-00296-f002:**
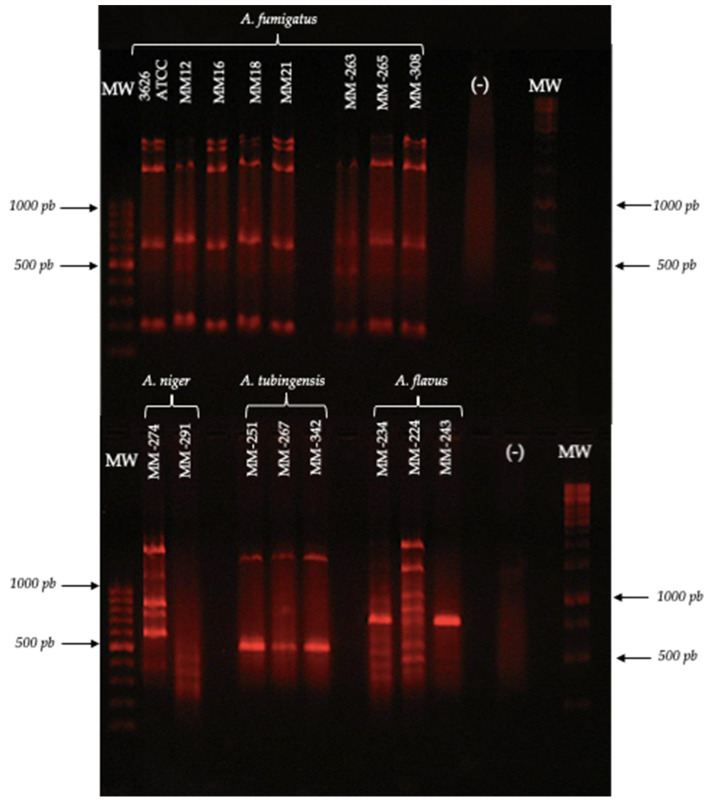
Polymorphic patterns were obtained by RAPD-PCR with the primer OPA-19. Conditions were as described in the Materials and Methods section.

**Figure 3 jof-08-00296-f003:**
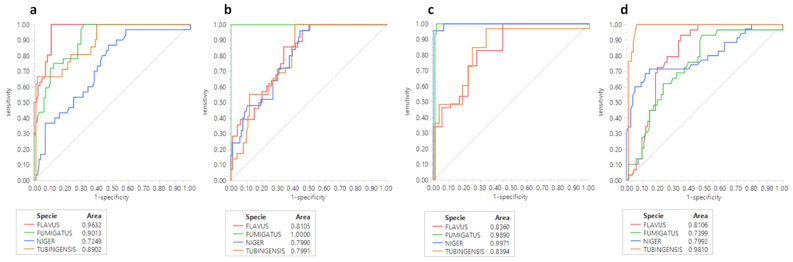
ROC curves of the quantitatively selected primers: (**a**) primer OPC-06; (**b**) primer OPF-01; (**c**) primer OPF-07 and (**d**) primer OPG-13. ROC-receiver operating characteristic.

**Figure 4 jof-08-00296-f004:**
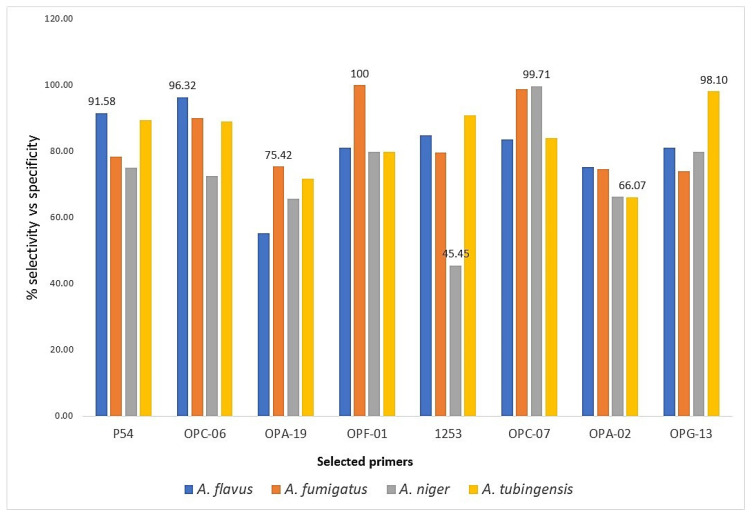
Percentage of sensitivity vs. specificity of the primers selected qualitatively (P54, OPA-19, 1253 and OPA-02) and quantitatively (OPC-06, OPF-01, OPF-07 and OPG-13) for each species of *Aspergillus* analyzed.

**Figure 5 jof-08-00296-f005:**
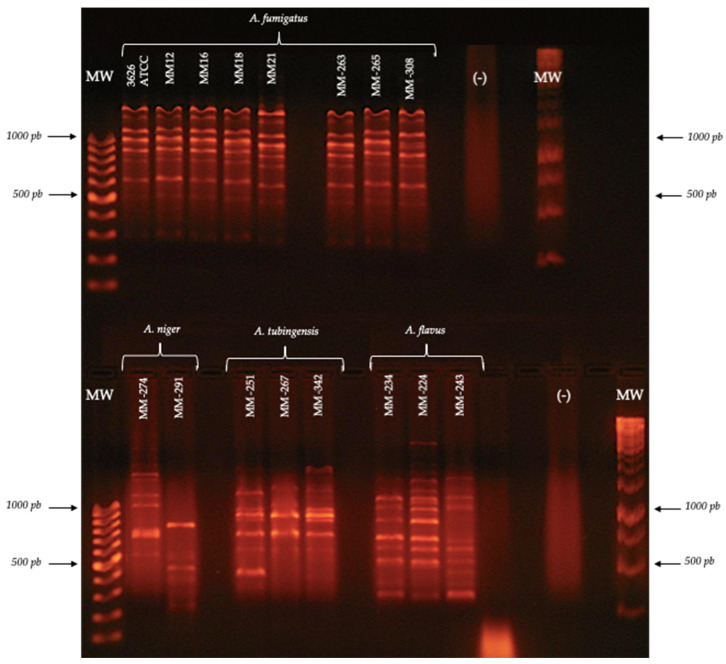
RAPD-PCR was performed for *A. fumigatus*, *A. niger*, *A. tubingensis* and *A. flavus* with the primer OPF-01.

**Table 1 jof-08-00296-t001:** Source and geographical origin of the isolates and reference strains.

Species	Isolate/Access Number	Source	Geographical Origin
*A. flavus*	9343D-2	ATCC	ND
	323C/OM89286	Environmental	MX
	87A/OM892858	Environmental	MX
	A40/OM892868	Environmental	MX
	370B/OM892869	Environmental	MX
	MM-234/MT347712.1	Environmental	CU
	MM-224/MT347711.1	Environmental	CU
	MM-243/MT347713.1	Environmental	CU
*A. fumigatus*	MYA3626	ATCC	ND
	MM12/OM892862	Clinical	AR
	MM16/OM89286	Clinical	AR
	MM18/OM892864	Clinical	AR
	MM21/OM892865	Clinical	AR
	MM-263/MT347701.1	Environmental	CU
	MM-265/MT347702.1	Environmental	CU
	MM-308/MT347703.1	Environmental	CU
*A. lentulus*	MYA3566	ATCC	ND
*A. niger*	WB326	ATCC	ND
	387A/OM892858	Environmental	MX
	39A/OM892859	Environmental	MX
	335B/OM892860	Environmental	MX
	88C/OM892861	Environmental	MX
	MM-274/MT410062.1	Environmental	CU
	MM-291/MT410063.1	Environmental	CU
*A. tubingensis*	1004	Reference strain	MX
	281C/OM892871	Environmental	MX
	146A/OM892872	Environmental	MX
	A26/OM892870	Environmental	MX
	220A/OM892873	Environmental	MX
	MM-251/MT410082	Environmental	CU
	MM-267/MT410083.1	Environmental	CU
	MM-342/MT410084.1	Environmental	CU

ATCC: American Type Culture Collection; ND: not known; MX: Mexico; AR: Argentina; CU: Cuba.

**Table 2 jof-08-00296-t002:** Reference sequences obtained from GenBank.

Species	Access Number
*A. niger*	MT410061.1 MT410062.1 MT410063.1
*A. tubingensis*	MT410082.1 MT410083.1 MT410083.1
*A. flavus*	MT347712.1 MT347711.1 MT347711.1
*A. fumigatus*	MT347701.1 MT347702.1 MT347703.1
*A. lentulus*	MN275504.1 MN275503.1 N275501.1

**Table 3 jof-08-00296-t003:** Primers used for the RAPD-PCR technique.

Primer	Sequence 5′-3′
1281	AACGCGCAAC
1283	CGGATCCCCA
R108	GTATTGCCCT
1253	GTTTCCGCCC
OPA-19	CAAACGTCGG
OPA-16	AGCCAGCGAA
OPB-12	CCTTGACGCA
OPA-17	GACCGCTTGT
OPA-15	TTCCGAACCC
OPF-05	CCGAATTCCC
OPA-08	GTGACGTAGG
OPC-07	GTCCCGATGA
OPG-01	CTACGGAGGA
OPG-03	GAGCCCTCCA
OPG-13	CTCTCCGCCA
OPG-15	ACTGGGACTC
OPC-06	GAACGGACTC
OPG-05	CTGAGACGGA
OPM-12	GGGACGTTGG
OPH-03	AGACGTCCAC
OPG-07	GAACCTGCGG
OPH-18	GAATCGGCCA
OPF-01	ACGGATTCTG
OPF-07	CCGATATCCC
OPF-09	CCAAGCTTCC
OPE-02	GGTGCGGGAA
P54	GGCGATTTTTGCCG
P160	CATGGCCACC
PELF	ATATCATCGAAGCCGC
B-04	TGCCATCAGT
OPH-17	CACTCTCCTC
T3B	AGGTCGCGGGTTCGAATCC
OPA-02	TGCCGAGCTG
OPA-03	AGTCAGCCAC

**Table 4 jof-08-00296-t004:** Species-specific primers qualitatively selected for each species.

Species-Specific Primers	Identified Species	Number of Bands Generated	Molecular Size (pb)
OPA-19	*A. fumigatus*	3–5	200–2400
1253	*A. niger*	13	190–16,002
OPA-02	*A. tubingensis*	2	500–1600
P54	*A. flavus*	13	190–2300

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
