# Peer review of "Selection of Polymorphic Patterns Obtained by RAPD-PCR through Qualitative and Quantitative Analyses to Differentiate Aspergillus fumigatus"

_jof, 2022, doi:10.3390/jof8030296_

Round 1
Reviewer 1 Report
Dear Dr(s)
Happy day.
The paper is very good.
Better to direct it to be more specific for A fumigatus which give the best result.
It might be that I have uploaded the original version.
Here you can find my revised file as attached.
My comment:
1- I am highly recommended that the title might be changed to be more focus on A. fumigatus while it give the best result according the representative data which is an impotent result.
2- The authors describe reference numbers in position that the author names is required. They advised to correct all of them through the paper.
3- All Latina number. symbol, words should be italic.
4- Over estimated words must be removed such as good.
5- Not all the used strains are described in the paper. The numbers should be agree in the text and in the tables. Strains that are not from the culture collection should given a number(s).
In general the paper needed only minor revision.

Author Response
Reviewer 1
Happy day.
The paper is very good.
Better to direct it to be more specific for A fumigatus which give the best result.
It might be that I have uploaded the original version.
Here you can find my revised file as attached.
My comment:
1- I am highly recommended that the title might be changed to be more focus on A. fumigatus while it give the best result according the representative data which is an impotent result.
Answer: The title has been changed as you suggested.
2- The authors describe reference numbers in position that the author names is required. They advised to correct all of them through the paper.
Answear: References cited in the manuscript have been corrected.
3- All Latina number. symbol, words should be italic.
Answear: They were corrected in the manuscript.
4- Over estimated words must be removed such as good.
Answer: The manuscript was corrected.
5- Not all the used strains are described in the paper. The numbers should be agree in the text and in the tables. Strains that are not from the culture collection should given a number(s).
Answer: Table 2 was corrected.
In general the paper needed only minor revision.
Reviewer 2 Report
The manuscript #jof-1593304, entitled “Selection of polymorphic patterns obtained by RAPD-PCR, through qualitative and quantitative analysis to differentiate species of Aspergillus spp.” by Valencia-Ledezma et al. presents a comprehensive manuscript on (in line with title) identification and differentiation of Aspergillus spp. using novel RAPD-PCR approach. The manuscript is very well prepared, edited and planned. I, personally, appreciate the fact that the Authors were very critical when discussing their results. Most of my issues are some minor editorial mistakes, examples below:
Lines 39 and 43: "Fumigati" and "Fumigati" should be unified
Captions of Tables/Figures should end with dot, ".".
The shortage of Aspergillus, "A." should be divided by space from the species name. Examples are in Table 4 - "A.fumigatus" should be "A. fumigatus"
Some of the figures are blurry, please provide a better resolution.
"Aspergillus species" should be "Aspergillus spp.", similarly in the title "... differentiate species of Aspergillus spp." should be "... differentiate Aspergillus spp." or "... differentiate species of Aspergillus genus".
Author Response
Reviewer 2
The manuscript #jof-1593304, entitled “Selection of polymorphic patterns obtained by RAPD-PCR, through qualitative and quantitative analysis to differentiate species of Aspergillus spp.” by Valencia-Ledezma et al. presents a comprehensive manuscript on (in line with title) identification and differentiation of Aspergillus spp. using novel RAPD-PCR approach. The manuscript is very well prepared, edited and planned. I, personally, appreciate the fact that the Authors were very critical when discussing their results. Most of my issues are some minor editorial mistakes, examples below:
Lines 39 and 43: "Fumigati" and "Fumigati" should be unified
Answer: They were corrected in the manuscript.
Captions of Tables/Figures should end with dot, ".".
Answer: The Tables and Figures were corrected.
The shortage of Aspergillus, "A." should be divided by space from the species name. Examples are in Table 4 - "A.fumigatus" should be "A. fumigatus"
Answer: They were corrected in the manuscript.
Some of the figures are blurry, please provide a better resolution.
Answer: The resolution of the figures has been improved.
"Aspergillus species" should be "Aspergillus spp.", similarly in the title "... differentiate species of Aspergillus spp." should be "... differentiate Aspergillus spp." or "... differentiate species of Aspergillus genus".
Answer: They were corrected in the manuscript.
Reviewer 3 Report
This paper describes selection of polymorphic patterns obtained by RAPD-PCR based on qualitative and quantitative analysis to differentiate species of Aspergillus spp. The paper is well written, but for acceptance major revisions is required. I have some comments here and also included in the manuscript file attached.
- In Figure 2. On the left side, author should write the approximate size of each DNA fragment (i.e. 500 bp, 1000 bp, etc).
- In phylogeny, I recommend to include type strains for the different species, A. fumigatus, A. niger, A. flavus, and A. tubingensis.
- Regarding Figure legend, it needs to include more information about phylogenetic analysis such as parameters, using GTR+G+I model? How many taxa? How long was the β-tubulin sequence?
- Lines 223-227: The information about phialide, metulae, and vescicle sizes should be added.
- Colony and microscopic pictures need to be inserted.
- Information about thirty-one fungal isolates such as sources, location, date, etc should be provided in Table 1.
- GenBank accession numbers of the β-tubulin sequences of thirty-one fungal isolates should be included.
- In Figure 2, polymorphic patterns were obtained by RAPD-PCR with the primer OPA-19 had two wells for the same isolate MM-224 of Aspergillus flavus. The author should explain it in Material and Methods section.
- Lines 395-398: It is not correct that Genbank does not have reference sequences of all the species of the section Fumigati. Most of species in section Fumigati (including type strains) were listed in Houbraken et al. (2020) (Please see: Studies in Mycology 2020;95: 5–169).

Author Response
Reviewer 3
This paper describes selection of polymorphic patterns obtained by RAPD-PCR based on qualitative and quantitative analysis to differentiate species of Aspergillus spp. The paper is well written, but for acceptance major revisions is required. I have some comments here and also included in the manuscript file attached.
- In Figure 2. On the left side, author should write the approximate size of each DNA fragment (i.e. 500 bp, 1000 bp, etc).
Answer: It was corrected in the manuscript.
- In phylogeny, I recommend to include type strains for the different species, A. fumigatus, A. niger, A. flavus, and A. tubingensis.
- Answer: The phylogenetic analysis included reference strains obtained from the ATCC (A. fumigatus, A. lentulus, A. niger, A. flavus, and A. tubingensis).
Regarding Figure legend, it needs to include more information about phylogenetic analysis such as parameters, using GTR+G+I model? How many taxa? How long was the β-tubulin sequence?
Answer: The requested information was included in the legend of Figure 1.
- Lines 223-227: The information about phialide, metulae, and vescicle sizes should be added.
- Colony and microscopic pictures need to be inserted.
- Answer: The requested information was included in the manuscript.
- Information about thirty-one fungal isolates such as sources, location, date, etc should be provided in Table 1.
Answer: The requested information was included in Table 1.
- GenBank accession numbers of the β-tubulin sequences of thirty-one fungal isolates should be included.
Answer: The access numbers of the Aspergillus isolates deposited in GenBank were included in Table 2, and the isolates marked with an asterisk are in the process of being registered in the Genbank with an id: 2556597.
- In Figure 2, polymorphic patterns were obtained by RAPD-PCR with the primer OPA-19 had two wells for the same isolate MM-224 of Aspergillus flavus. The author should explain it in Material and Methods section.
Answer: Figure 2 and 3 were edited to remove duplicate lanes, which corresponded to the same isolate, to avoid confusion.
- Lines 395-398: It is not correct that Genbank does not have reference sequences of all the species of the section Fumigati. Most of species in section Fumigati (including type strains) were listed in Houbraken et al. (2020) (Please see: Studies in Mycology 2020;95: 5–169).
Answer: We appreciate your observation and we agree with you, so we have eliminated the incorrect information.
Round 2
Reviewer 3 Report
It is acceptable now. Congratulations!